# DISCOVERING BUGS IN VISION MODELS USING OFF-THE-SHELF IMAGE GENERATION AND CAPTIONING

## ABSTRACT

Automatically discovering failures in vision models under real-world settings remains an open challenge. This work shows how off-the-shelf, large-scale, image-to-text and text-to-image models, trained on vast amounts of data, can be leveraged to automatically find such failures. In essence, a conditional text-to-image generative model is used to generate large amounts of synthetic, yet realistic, inputs given a ground-truth label. A captioning model is used to describe misclassified inputs. Descriptions are used in turn to generate more inputs, thereby assessing whether specific descriptions induce more failures than expected. As failures are grounded to natural language, we automatically obtain a high-level, human-interpretable explanation of each failure. We use this pipeline to demonstrate that we can effectively interrogate classifiers trained on IMAGENET to find specific failure cases and discover spurious correlations. We also show that we can scale the approach to generate adversarial datasets targeting specific classifier architectures. This work demonstrates the utility of large-scale generative models to automatically discover bugs in vision models in an open-ended manner. We also describe a number of limitations and pitfalls related to this approach.

## 1 INTRODUCTION

Deep learning has enabled breakthroughs in a wide variety of fields (Goodfellow et al., 2016; Krizhevsky et al., 2012; Hinton et al., 2012), and deep neural networks are ubiquitous in many applications, including autonomous driving (Bojarski et al., 2016) and medical imaging (De Fauw et al., 2018). Unfortunately, these models are known to exhibit numerous failures arising from using *shortcuts* and *spurious correlations* (Geirhos et al., 2020a; Arjovsky et al., 2019; Torralba et al., 2011; Kuehlkamp et al., 2017). As a result, they can fail catastrophically when training and deployment data differ (Buolamwini & Gebru, 2018). Hence, it is important to ensure that models are robust and generalize to new deployment settings.

Yet, only a few tools exist to automatically find failure cases on unseen data. Some methods analyze the performance of models by collecting new datasets (usually by scraping the web). These datasets must be large enough to obtain some indication of how models perform on a particular subset of inputs (Hendrycks et al., 2019; 2020; Recht et al., 2019). Other methods rely on expertly crafted, synthetic (and often unrealistic) datasets that highlight particular shortcomings (Geirhos et al., 2022; Xiao et al., 2020).

In this work, we present a methodology to automatically find failure cases of image classifiers in an open-ended manner, without prior assumptions on the types of failures and how they arise. We leverage off-the-shelf, large-scale, text-to-image, generative models, such as DALL·E 2 (Ramesh et al., 2022), IMAGEN (Saharia et al., 2022) or STABLE-DIFFUSION (Rombach et al., 2022), to obtain realistic images that can be reliably manipulated using the text prompt. We also leverage captioning models, such as FLAMINGO (Alayrac et al., 2022) or LEMON (Hu et al., 2021), to retrieve human-interpretable descriptions of each failure case. This provides the following advantages: *(i)* generative models trained on web-scale datasets can be re-used and have broad non-domain-specific coverage; *(ii)* they demonstrate basic compositionality, can generate novel data and can faithfully capture the essence of (most) prompts, thereby allowing images to be realistically manipulated; *(iii)* textual

descriptions of failures can be easily interpreted (even by non-experts) and interrogated (e.g., by performing counterfactual analyses). Overall, our contributions are as follows:

- We describe a methodology to discover failures of image classifiers trained on IMAGENET (Deng et al., 2009). To the contrary of prior work, we leverage off-the-shelf generative models, thereby avoiding the need to collect new datasets or to rely on manually crafted synthetic images.

- Our approach surfaces failures that are human-interpretable by clustering and captioning inputs on which classifiers fail. These captions can be modified to produce alternative hypotheses of why failures occur allowing insights into the limitations of a given model.

- We demonstrate the scalability of the approach by generating adversarial datasets (akin to IMAGENET-A; Hendrycks et al., 2019). In contrast to IMAGENET-A, our new generated datasets align more closely with the original training distribution from IMAGENET and generalize to multiple classifier architectures.

Importantly, while this work focuses on vision models trained on IMAGENET, it is neither limited to IMAGENET nor the visual domain. It serves as a **proof-of-concept** that demonstrates how large-scale, off-the-shelf, generative models (Bommasani et al., 2021) can be combined to automate the discovery of *bugs* in machine learning models and produce compelling, interpretable descriptions of model failures. The approach is agnostic to the model architecture, which can be treated as a black box.

## 2 RELATED WORK

**Model failures.** Spurious correlations can entice models to learn unintended shortcuts that obtain high accuracy on the training set but fail to generalize (Lapuschkin et al., 2019; Geirhos et al., 2020a). Recht et al. (2019) show that the accuracy of IMAGENET models is impacted by changes in the data collection process, while Torralba et al. (2011); Khosla et al. (2012); Choi et al. (2012) explore how contextual bias affects generalization. Mania et al. (2019) demonstrate that models trained on IMAGENET make consistent mistakes with one another and Geirhos et al. (2020b) show that these mistakes are not necessarily consistent with human judgment.

**Evaluation datasets.** Understanding how model failures arise and empirically analyzing their consequences often requires collecting and annotating new test datasets. Hendrycks et al. (2019) collected datasets of natural adversarial examples (IMAGENET-A and IMAGENET-O) to evaluate how model performance degrades when inputs have limited spurious cues. Hendrycks et al. (2020) collected four real-world datasets (including IMAGENET-R) to understand how models behave under distribution shifts. In many cases, particular shortcomings can only be explored using synthetic datasets (Cimpoi et al., 2013). Hendrycks & Dietterich (2018) introduced IMAGENET-C, a synthetic set of common corruptions. Geirhos et al. (2018) propose to use images with a texture-shape cue conflict to evaluate the propensity of models to over-emphasize texture cues. Xiao et al. (2020); Sagawa et al. (2020) investigate whether models are biased towards background cues by compositing foreground objects with various background images (IMAGENET-9, WATERBIRDS). In all cases, building such datasets is time-consuming and requires expert knowledge.

**Automated failure discovery.** In some instances, it is possible to distill rules or specifications that constrain the input space enough to enable the automated discovery of failures via optimization or brute-force search. In vision tasks, adversarial examples, which are constructed using $\ell_p$-norm bounded perturbations of the input, can cause neural networks to make incorrect predictions with high confidence (Carlini & Wagner, 2017a;b; Goodfellow et al., 2014; Kurakin et al., 2016; Szegedy et al., 2013). In language tasks, some efforts manually compose templates to generate test cases for specific failures (Jia & Liang, 2017; Garg et al., 2019; Ribeiro et al., 2020). Such approaches rely on human creativity and are intrinsically difficult to scale. Several works (Baluja & Fischer, 2017; Song et al., 2018; Xiao et al., 2018; Qiu et al., 2019; Wong & Kolter, 2021; Laidlaw et al., 2020; Gowal et al., 2019) go beyond hard-coded rules by leveraging generative and perceptual models. However, such approaches are difficult to automate as it is unclear how to relate specific latent variables to isolated structures of the original input. Finally, we highlight a concurrent work (Ge et al., 2022), which leverages captioning and text-to-image models to construct background images to evaluate (and improve) an object detector. Their approach requires compositing the resulting images with foreground objects and is not open-ended, in the sense that it requires a dataset of background images. Perhaps, the work by Perez et al. (2022) on *red-teaming* language models is the most similar to ours.

Perez et al. demonstrate how to prompt a language model to automatically generate test cases to probe another language model for toxic and other unintended output.

**Interpretability.** Interpretability techniques aim to explain a rationale behind individual model predictions. They surface information that allows a user to inspect how a model behaves. Casper et al. (2022) demonstrate that feature-level attacks, which create adversarial patches, can help diagnose brittle feature associations. However, much like LIME (Ribeiro et al., 2016) or GRAD-CAM (Selvaraju et al., 2019), the results are difficult to understand. Other works (Abid et al., 2022; Jain et al., 2022a; Eyuboglu et al., 2022) leverage auxiliary information in the form of attributes or image-to-text embeddings (e.g., from CLIP; Radford et al., 2021) to provide explanations in natural language. However, these methods often rely on an additional dataset which limits their scope.

## 3 METHOD

**Notation.** We consider a classifier $f : \mathbb{X} \to \mathbb{Y}$, where $\mathbb{X}$ is the set of inputs (i.e., images) and $\mathbb{Y}$ is the label set. We also assume that inputs $\boldsymbol{x} \in \mathbb{X}$ with label $y \in \mathbb{Y}$ are drawn from an underlying distribution $p(\mathbf{x}|\boldsymbol{z}, y)$ conditioned on latent representations $\boldsymbol{z} \in \mathbb{Z}$. In the context of this specific work, $\boldsymbol{z}$ is a textual description of the image $\boldsymbol{x}$. We are interested in discovering captions $\boldsymbol{z}$ corresponding to images $\boldsymbol{x} \sim p(\mathbf{x}|\boldsymbol{z}, y)$ with label $y$ that lead to significantly higher misclassification rates than generic images drawn from the marginal distribution $p(\mathbf{x}|y)$ conditioned solely on the label. Formally, given a label $y$, we would like to find $\boldsymbol{z}$ with

$$\mathbb{E}_{\boldsymbol{x} \sim p(\mathbf{x}|\boldsymbol{z}, y)} [f(\boldsymbol{x}) \neq y] > \mathbb{E}_{\boldsymbol{x} \sim p(\mathbf{x}|y)} [f(\boldsymbol{x}) \neq y] \tag{1}$$

where $[\cdot]$ represents the Iverson bracket. We may also be interested in identifying specific misclassifications towards a wrong (target) label $\bar{y} \neq y$, and would like that

$$\mathbb{E}_{\boldsymbol{x} \sim p(\mathbf{x}|\boldsymbol{z}, y)} [f(\boldsymbol{x}) = \bar{y}] > \mathbb{E}_{\boldsymbol{x} \sim p(\mathbf{x}|y)} [f(\boldsymbol{x}) = \bar{y}]. \tag{2}$$

As we do not have access to the true underlying distributions $p(\mathbf{x}|\boldsymbol{z}, y)$ and $p(\mathbf{x}|y)$, we leverage a large-scale text-to-image model (such as DALL·E 2 or IMAGEN) to approximate them. Similarly, we approximate $p(\mathbf{z}|\boldsymbol{x}, y)$ with a captioning model (e.g., FLAMINGO or LEMON). We denote approximations of these distributions with the symbol $\hat{p}$.

> For each of the following steps, we highlight additional implementation details and explain how we construct prompts for the text-to-image and image-to-text models. Some minute details are omitted as to not break anonymity and will be added later.

**Generating failure cases.** Our approach is described in Fig. 1. It consists of initially finding *baseline* failures for the underlying model $f$ by sampling inputs $\boldsymbol{x}$ from $\hat{p}(\mathbf{x}|y)$.[1] Given a label of interest $y$, the output of this step is a set $\mathbb{D} = \{\boldsymbol{x}_i \sim \hat{p}(\mathbf{x}|y)\}_{i=1}^{N}$ (where $N$ is the number of images we wish to generate), a set $\mathbb{D}_{\text{fail}} = \{\boldsymbol{x} \in \mathbb{D}|f(\boldsymbol{x}) \neq y\}$ and an estimate of the baseline failure rate $|\mathbb{D}_{\text{fail}}|/N$ (corresponding to the right-hand side of Eq. 1). In this work, we consider problematic misclassifications only so we restrict ourselves to failures where any of the top-3 predicted labels are not under the same parent as the true label $y$ in the WORDNET hierarchy (Miller, 1995).[2]

> This step leverages a text-to-image generative model conditioned on $y$ with prompts such as *"a realistic photograph of a fly (insect)."*, which are automatically generated from the corresponding class and WORDNET hierarchy. As our domain of interest is composed of real images (much like the images in IMAGENET), the prompt is designed to enforce the generation of photographs of real objects and animals rather than drawings or paintings. Further prompt engineering can be explored to find failures on a wide variety of domains (such as sketches or medical imaging; Hendrycks et al., 2020; Kather et al., 2022).

---

[1]In practice, it is possible to steer the generation process to induce more frequent failures (e.g., by optimizing latents or conditioning via gradient ascent on the cross-entropy loss; Wong & Kolter, 2021). However, as a proof-of-concept, we assume only black-box access to off-the-shelf text-to-image model.

[2]Other options include considering only misclassifications where the true label is not in the top-$k$ predicted labels or where the confidence in the true label is lower than a predefined threshold.

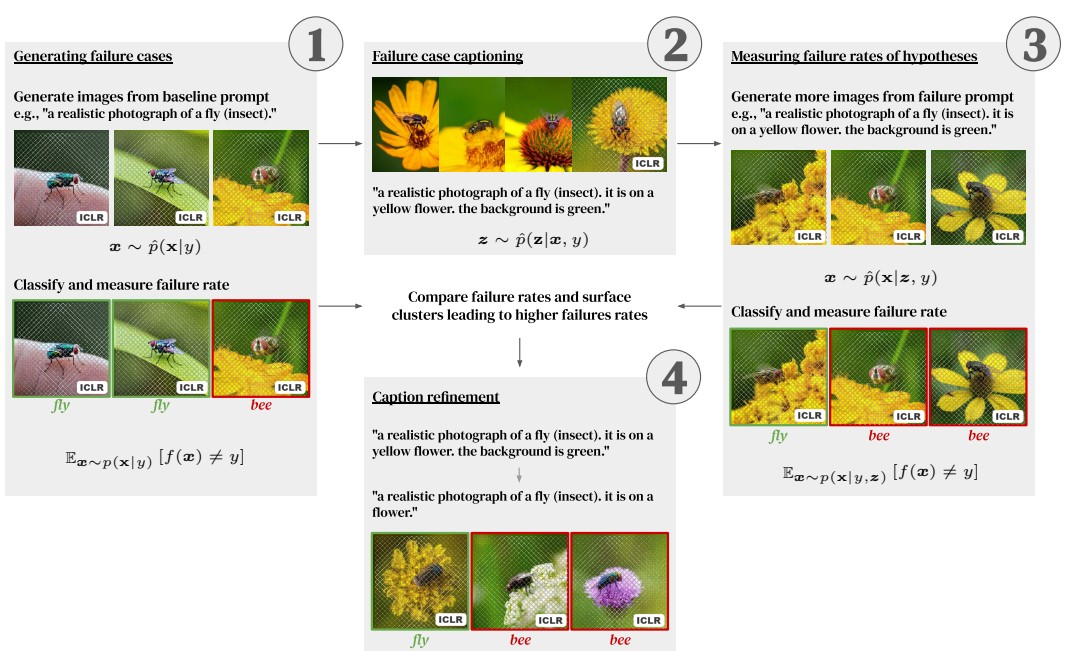

**Figure 1: Diagram of our method.** The method starts by generating images containing a given class $y$ to measure the baseline failure rate of that class (right-hand side of Eq. 1). We construct a textual description for each misclassified image. This description is used to produce new images and measure the failure rate on images corresponding to that description (left-hand side of Eq. 1). The final description can be edited (manually or automatically) to understand the source of the failures.

**Optional clustering failure cases.** Clustering is not necessary, but makes the search for the caption $z$ leading to high failure rates more efficient and computationally manageable. First, we split $\mathbb{D}_{\text{fail}}$ by predicted label, i.e. $\mathbb{D}_{\text{fail}} = \mathbb{D}_{\text{fail}}^{(1)} \cup \ldots \cup \mathbb{D}_{\text{fail}}^{(|\mathcal{Y}|)}$, where $\mathbb{D}_{\text{fail}}^{(y)} = \{\boldsymbol{x} \in \mathbb{D}_{\text{fail}} | f(\boldsymbol{x}) = y\}$. Then, for each subset $\mathbb{D}_{\text{fail}}^{(y)}$, we group inputs that have similar feature representations together (e.g., using the cosine distance between intermediate activations of a pretrained model). Many other choices of clustering method exist. The goal of this step is to reduce the number of clusters to a minimum without amalgamating different causes of failure together.[3]

**Failure case captioning.** For each cluster $\mathbb{A} \subseteq \mathbb{D}_{\text{fail}}$, we would like to find a caption $\boldsymbol{z}_{\mathbb{A}}$ that describes it. Grounding failures in simple textual descriptions allows us to maintain the diversity of the generated images: the generated images resemble the images leading to the original failure without being exact copies. Ideally, we would like to find the caption $\boldsymbol{z}_{\mathbb{A}}$ that maximizes the likelihood of sampling elements of $\mathbb{A}$, i.e., $\boldsymbol{z}_{\mathbb{A}} = \arg\max_{\boldsymbol{z}} \prod_{\boldsymbol{x} \in \mathbb{A}} \hat{P}(\boldsymbol{x}|\boldsymbol{z}, y)$. We may wish to impose constraints on $\boldsymbol{z}_{\mathbb{A}}$, such as a maximum number of words or sentences.[4] Finding such a caption is computationally hard and measuring exact likelihoods $\hat{P}(\boldsymbol{x}|\boldsymbol{z}, y)$ can be challenging. Hence, we resort to sampling captions directly from a captioning model $\hat{p}(\boldsymbol{z}|\boldsymbol{x}, y)$ for each image of cluster $\mathbb{A}$.[5] Captions are split into sentences, resulting in a set of sentences $\mathbb{S}$. Sentences are greedily combined (up to a maximum number of sentences $K$) to maximize the likelihood of sampling the overall caption $\boldsymbol{z}_{\mathbb{A}} = [\boldsymbol{s}_1, \ldots, \boldsymbol{s}_K]$ with $\boldsymbol{s}_j = \arg\max_{\boldsymbol{s} \in \mathbb{S}} \prod_{\boldsymbol{x} \in \mathbb{A}} \hat{P}([\boldsymbol{s}_1, \ldots, \boldsymbol{s}_{j-1}, \boldsymbol{s}]|\boldsymbol{x}, y)$ and $\boldsymbol{s}_1 = \arg\max_{\boldsymbol{s} \in \mathbb{S}} \hat{P}([\boldsymbol{s}]|\boldsymbol{x}, y)$. If no clustering is performed, we can directly sample a caption $\boldsymbol{z}_{\{\boldsymbol{x}\}} \sim \hat{p}(\boldsymbol{z}|\boldsymbol{x}, y)$ from the captioning model for each element $\boldsymbol{x}$ in $\mathbb{D}_{\text{fail}}$. Each caption serves as a failure hypothesis.

---

[3]Wrongful clustering may lead the next step to produce descriptions that fail to induce more failures. As a result, we may miss failures that we would have discovered without clustering, but the failures that we do discover remain valid.

[4]These constraints guarantee that captions remain simple and not overly descriptive.

[5]This formulation implicitly assumes that any caption is as likely as another under a given label $y$, which in general does not hold true, but serves as a reasonable approximation.

> This step leverages an image-to-text captioning model and forces descriptive completions via few-shot prompting. In particular, we use three shots, with each shot being an image and caption pair. Images are publicly available online and are manually described using between four to seven short sentences that describe the subject of the photograph and its physical position with respect to the camera, as well as the background or context in which the subject appears (see Appendix A for more details). To condition on $y$, we force the captioning model to only consider completions to the original baseline prompt, which guarantees that the final caption contains the true label. An example of a resulting caption is *"a realistic photograph of a fly (insect). the background is blurred. the fly is in focus. it is on a yellow flower. the background is green."* Each caption serves as a failure hypothesis.

**Measuring failure rates of hypotheses.** For each failure hypothesis or caption $z_{\mathbb{A}}$, we can measure its failure rate via sampling $\mathbb{E}_{\boldsymbol{x} \sim p(\mathbf{x} | z_{\mathbb{A}}, y)} [f(\boldsymbol{x}) \neq y]$. This step allows us to surface captions $z^{\star}$ that satisfy Eq. 1 (or alternatively Eq. 2) by comparing the resulting failure rate with the baseline failure rate obtained initially.

> Much like the first step on generating the baseline failure cases, this step uses the text-to-image model, which we now prompt with descriptive cluster captions such as *"a realistic photograph of a fly (insect). the background is blurred. the fly is in focus. it is on a yellow flower. the background is green."*

**Caption refinement and counterfactual analysis.** Given a caption $z^{\star}$, we would like to provide a shorter, self-contained caption that obtains a similar failure rate. For this step, we rely on simple rules[6] and evaluate promising caption rewrites. Finally, as captions are human-readable, users can interact with the system and test alternative hypotheses.

> In our implementation, we exploit two rules: *(i)* we evaluate all individual sentences (for our ongoing example, these would be *"the background is blurred"*, *"the fly is in focus"*, *"it is on a yellow flower"* and *"the background is green"*) in conjunction with the original prompt (e.g., *"a realistic photograph of a fly (insect)"*); *(ii)* the most promising prompt is further refined by dropping adjectives (such as *"yellow"* in *"it is on a yellow flower"*). More sophisticated rules and rewrites are possible (Ribeiro et al., 2018). Alternatively, we could leverage a large-language model via few-shot prompting to automate this process (Witteveen & Andrews, 2019).

Finally, we note that it is possible to use part of this pipeline to understand known failure cases (e.g., on failures reported by external users of the model $f$).

## 4 RESULTS

In this section, we elaborate on two use-cases. In the first, we find failure cases of a Residual Network (RESNET) (He et al., 2016) model with 50 layers (RESNET-50), trained on IMAGENET, in an open-ended manner. We focus on three arbitrarily chosen labels and show that the failures we obtain arise from consistent misclassifications caused by spurious correlations. In the second use-case, we show that we can generate failures at scale for various models. We leverage IMAGENET-A as a source of initial failures and create new variants of IMAGENET-A that target specific model architectures.

### 4.1 OPEN-ENDED FAILURE SEARCH

**Setup.** We evaluate a RESNET-50 trained on IMAGENET and available on TF-HUB.[7] We select three labels $y$ at random: `Persian cat`, `fly` and `crayfish`. For each label $y$, we manually select target labels $\bar{y}$ (`snow leopard`, `bee` and `chainlink fence` respectively) and execute the protocol defined in Sec. 3. At each step, we sample images from the generative model until we gather 20 images that are misclassified as the target label $\bar{y}$ and compute failure rates at that point. Fig. 2 and Table 1 show these automatically discovered failures. More failures for additional true and

---

[6]This may seem at odds with our claim on open-endedness. However, we note that this step is optional and its goal is simply to produce shorter failure descriptions.

[7]https://tfhub.dev/google/imagenet/resnet_v2_50/classification/5

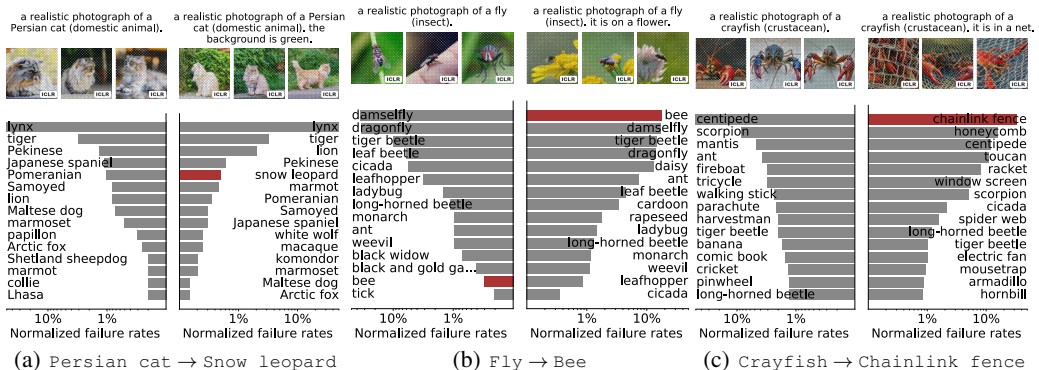

**Figure 2: Distribution of failures of a RESNET-50 for the baseline and automatically discovered captions for three true and target label pairs.** For each panel, failures resulting from the baseline caption are on the left and failures resulting from the discovered caption are on the right. We show the top-15 mistakes and three randomly sampled images for each caption. We highlight in red the bar corresponding to the target label. Absolute failure rates are given in Table 1.

| True label | Target label | Caption | Failure rate (any) | | Failure rate (target) | |
|---|---|---|---|---|---|---|
| Persian cat | Snow leopard | a realistic photograph of a Persian cat (domestic animal). | 0.11% | 1× | 0.00022% | 1× |
| | | — " — the background is green. | 0.64% | 6× | 0.0032% | 14× |
| Fly | Bee | a realistic photograph of a fly (insect). | 0.48% | 1× | 0.0014% | 1× |
| | | — " — it is on a flower. | 4.11% | 9× | 0.72% | 497× |
| Crayfish | Chainlink fence | a realistic photograph of a crayfish (crustacean). | 0.93% | 1× | 0.00047% | 1× |
| | | — " — it is in a net. | 6.31% | 7× | 1.73% | 3721× |

**Table 1: Absolute failure rates of a RESNET-50 for three true and target label pairs.** We show the total failure rate (i.e., the model prediction is different from the true label) and the target failure rate (i.e., the model prediction is the target label). Captions are automatically discovered using the method detailed in Sec. 3.[8]

target label pairs are in Sec. B.1 in the appendix. In Sec. B.1, we also evaluate other architectures (i.e., VITs) and demonstrate that the discovered captions yield failure rates that are statistically significant.

**Discovered failures.** Fig. 2(a) shows the distribution of failures for the baseline label Persian cat. We observe that the most frequent confusion, on images generated using the baseline caption *"a realistic photograph of a Persian cat (domestic animal)."* is with lynx. This mistake arises about 0.1% of the time and constitutes 87.3% of all failures. In comparison, the confusion with snow leopard is rather infrequent and arises only 0.00022% of the time. Our approach automatically discovers that adding *"the background is green."* to the caption results in a large increase in failure rates. Failures are 5.72× more likely and the model is 14.3× more likely to predict snow leopard. We generally observe that mistakes with wild animals become more prevalent when the cat is outdoors. Similarly, Fig. 2(b) and Fig. 2(c) show failures on images of flies and crayfish, respectively. Flies on flowers are significantly more likely to be confused for bees when they are on flowers (497×), while crayfish in nets are more frequently confused as chainlink fences (3721×), honeycomb, window screens or spider webs. These highlight two shortcomings of the underlying classifier: *(i)* the over-reliance on spurious cues (such as the flower); *(ii)* the inability to determine which object is the main subject of a photograph (e.g., which of the net or crayfish is important).

**Generalizability of failure descriptions.** To verify that the discovered failures are not specific to the text-to-image model used in this manuscript and do not result from artifacts in the image generation process, we generate 30 images using the baseline and discovered captions with DALL·E 2 and STABLE-DIFFUSION (samples are shown in Fig. 13 and Fig. 14 in the appendix). We evaluate

---

[8] As a point of comparison, we can also evaluate the baseline failure rates on images from the IMAGENET test set. For Persian cat, fly and crayfish the baseline failure rates are 16%, 8% and 18%, respectively (the target failure rate is 0% for all labels). These failure rates are higher than on images generated by the generative model, this is perhaps indicating that the generative model produces images that are more canonical and conservative. We also point out that we generate 9.1M samples to compute the first row of the table, 625K for the second and 1.4M, 2.8K, 4.3M, 1.2K for the subsequent rows.

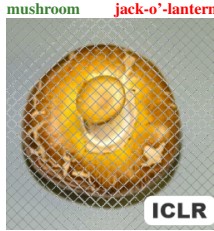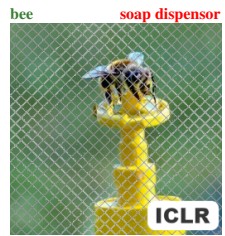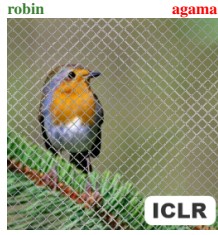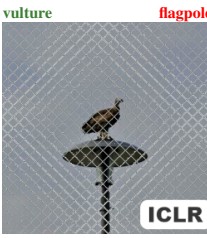

**Figure 3:** Examples of failures automatically found in IN-G-RN. The correct label is to the left in **green**. The incorrect prediction is to the right in **red**. Images are not watermarked upon classification.

the failure rates for the `fly` and `crayfish` labels (which exhibited higher failure rates). With DALL·E 2, for the 30 images generated with the prompt *"a realistic photograph of a fly (insect)."*, 18 are correctly classified as flies and none are classified as bees. When adding *"it is on a flower."* to the prompt, the overall failure rate increases (only 14 images are correctly classified) and nine images are now classified as bees. Similarly, for *"a realistic photograph of a crayfish (crustacean)."*, 29 images are correctly classified as either `crayfish`, `spiny lobster`, `American lobster`, `Dungeness crab` or `king crab`, while none are classified as `chainlink fence`. When adding *"it is in a net."*, four are classified as `chainlink fence` (with `chainlink fence` appearing ten times in the top-3 predictions), while only 21 images are correctly classified. Results are similar for STABLE-DIFFUSION images.[9] Overall, we observe that discovered failures generalize across generative models. Finally, we also leverage *Google Image Search*[10] to find 30 images for each of the following queries: *"fly"*, *"fly on flower"*, *"crayfish"*, *"crayfish in net"* (images must have a resolution of at least $256 \times 256$ and should contain the true label). We classify all images and observe that the number of failures towards `bee` increases from zero to two and those towards `chainlink fence` increase from zero to four. This illustrates again that discovered failures are general and extend to real photographs.

## 4.2 ADVERSARIAL DATASET GENERATION AT SCALE

We demonstrate how to apply our automated pipeline to generate large datasets of failures. We seed our search by captioning images from IMAGENET-A. We show that the discovered failures generalize across initializations of a given model architecture and between models of different architectures.

**Generating large-scale datasets.** We assume that we have access to a set of captions that describe potential failure cases.[11] These captions are automatically extracted from the 7,500 images of IMAGENET-A using the captioning model, limiting its output to two sentences maximum. For each caption, we sample up to a thousand images keeping those leading to misclassifications and limit the number of misclassified images kept per caption. We consider two models: a RESNET-50 (abbreviated hereafter by RN) and a Vision Transformer in its B/16 variant (Dosovitskiy et al., 2020), abbreviated by VIT. Both models are trained solely on IMAGENET and achieve 76% and 80% top-1 accuracy, respectively. This yields two separate datasets of failures which we refer to as IN-G-RN and IN-G-VIT that are of size 12,332 and 9,536 respectively.

**Visualizations of failure cases.** Samples from IN-G-RN are shown in Fig. 3, with true and predicted labels. We can see that while the images clearly show the correct class, the model erroneously predicts a different one. Additional samples for both IN-G-RN and IN-G-VIT are visible in Fig. 15 and Fig. 16 (in the appendix). Appendix C provides an analysis for some failure cases found in IN-G-VIT and details how images from the IMAGENET training set may have misled the classifier.

**Generalizability of failure cases.** To investigate whether the discovered failures generalize across initializations, we train an additional RESNET (RN2) and VIT (VIT2) on IMAGENET with the exact

---

[9]For STABLE-DIFFUSION, the number of failures increases from one to two and from four to 16, for the `fly` and `crayfish` labels, respectively.

[10]https://images.google.com

[11]This assumption is not necessary. However, it accelerates our search by generating images that are more likely to induce failures.

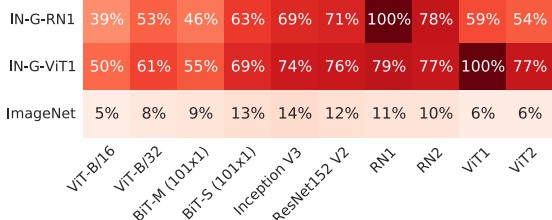

| Dataset | FID ↓ | KID ↓ |
|---|---|---|
| IMAGENET-A | 56.6 | 0.0460 |
| IN-G-RN | 48.3 | 0.0305 |
| IN-G-VIT | 53.9 | 0.0330 |
| IMAGENET (train) | 2.3 | 0.0003 |

**Figure 4: Failure rates (top-3) for different models on two generated datasets and IMAGENET.** We report the failure rates of different models trained on IMAGENET.

**Table 2: FID and KID scores.** We report FID and KID scores in relation to IMAGENET (test).

same setup as our two original models but different random seeds. We also consider a large set of additional models trained on IMAGENET and optionally pre-trained on larger datasets obtained from TF-HUB. Fig. 4 shows the failure rates induced by both datasets on all models (failures are accounted when the top-3 predictions do not include the true label). First, we observe that failures transfer well between models of the same architecture. Indeed, 78% of the failures in IN-G-RN transfer to RN2, while the ones in IN-G-VIT transfer with 77% chance to VIT2. Second, we observe that failures for a given model architecture transfer to a large extent across architectures. Even when large-scale pretraining is used (with the BIT-M (101x1), VIT-B/16 and VIT-B/32 models pretrained on IMAGENET21K), failures transfer at a rate of 39-53% for IN-G-RN and 50-61% for IN-G-VIT. Further results for additional models are in Sec. B.2 in the appendix.

**Distribution shift.** Finally, we compare our generated datasets to IMAGENET and IMAGENET-A. The aim is to validate that we generate images that are similar to those in IMAGENET. We compute the Fréchet Inception Distance (FID; Heusel et al., 2017) and the Kernel Inception Distance (KID; Binkowski et al., 2018) between the generated images and the IMAGENET test set. Table 2 also shows the FID and KID of the IMAGENET train set and IMAGENET-A. We find that our generated images are *more* similar to those from IMAGENET under both metrics than those from IMAGENET-A.

## 5 DISCUSSION AND LIMITATIONS

The motivation behind our work is to develop a proof-of-concept demonstrating that today's large-scale text-to-image and image-to-text models can be leveraged to find human-interpretable failures in vision models. While we focus exclusively on IMAGENET, there are encouraging signs that these generative models could be used to probe models trained on specialized tasks such as medical imaging (Kather et al., 2022). Overall, there remain a number of key challenges to address.

**Coverage.** While our approach can successfully be used to demonstrate the presence of failures, it is important to understand that (just like scraping the web) it cannot prove their absence. In other words, there is no guarantee that it will discover all failures of a given model.[12] Moreover, the generative model is only an approximation of the distribution of interest and may lack coverage. For example, it might almost never generate "a lawnmower falling down from the sky" (an actual image from the IMAGENET training set; Jain et al., 2022b) when prompted with *"a realistic photograph of a lawnmower"*. While this can help ground failures to scenes that are likely to occur in the real-world, it also means that rare failures are unlikely to be discovered (see Fig. 5(a)).

**Bias.** While we take the view here that off-the-shelf large-scale generative models are trained on diverse and unbiased data, the reality is far from it: these models mirror the distribution of images and captions seen on the web. The generative model may over-sample particular regions of the image manifold and, as a result, our approach is more likely to discover failures in these high-density regions and miss failures pertaining to other regions (see Fig. 5(b)). Possible solutions to reduce bias include clever prompting (which introduces expert knowledge) or discovering failure prompts more actively by avoiding random sampling (e.g., through adversarial techniques).

---

[12]However, if the generative model matches the true distribution of images, it is possible to provide meaningful probabilistic guarantees.

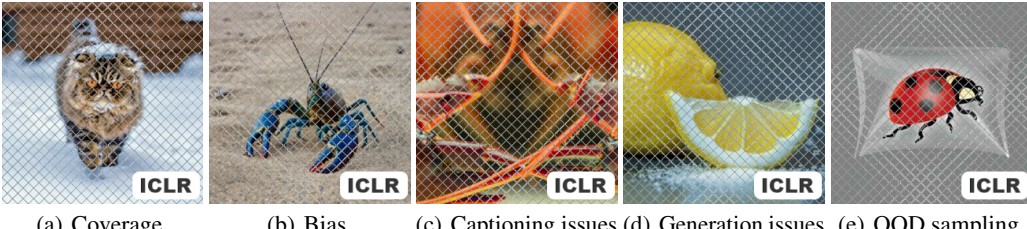

(a) Coverage     (b) Bias     (c) Captioning issues (d) Generation issues (e) OOD sampling

**Figure 5: Illustrative examples of various challenges.** (a) Persian cats in snow (generated using *"a realistic photograph of a Persian cat (domestic animal). it is walking in the snow."*) are misclassified as snow leopards at a rate of 0.016%, which is significantly higher than the failure rate of 0.0032% induced by the automatically found caption (*"— " — the background is green."*); the total failure rate also increases twelve-fold to 8.15% (from 0.64%). (b) It is estimated that only 1 in 10,000 crayfish turn blue. However, 9% of the images generated using *"a realistic photograph of a crayfish (crustacean)."* contain a blue crayfish (estimated by manually looking at 100 samples). (c) This image of a crayfish is misclassified as a chainlink fence. The output of the captioning model for this particular image is *"a realistic photograph of a crayfish. the crayfish is very detailed. the crayfish is facing the camera. the crayfish is orange. it has two antennae."* While the caption describes the image, it does not provide enough detail to reconstruct the image. (d) This image is generated from the caption *"a realistic photograph of a saltshaker (container). there is a lemon slice on the side of the salt shaker."* While the image contains a lemon, the true class $y$ (saltshaker) is not visible. (e) Generated with the caption *"a realistic photograph of a ladybug (insect). it is in a plastic bag."*, this image illustrates that text-to-image models can create image that are not from the intended distribution (i.e., of realistic photographs).

**Captioning issues.** Using captions as our latent representation allows our approach to produce human-interpretable explanations and constrains our search to failures that can be explained in words. Not only is it possible for the captioning model to miss important details or produce ungrounded captions, but some failures may simply be hard to describe (even by a human). As a result, newly generated images may look different from the set of images that induced the original failure. We note that efficiently enforcing consistency between the generated and original images (through a common caption) is an open problem since we would like to search over *reasonable* captions that are likely to produce images corresponding to the original failure. Fig. 5(c) shows an example. The figure shows a crayfish misclassified as a chainlink fence. While the reason for that failure is immediately obvious to us, it remains difficult to describe with a succinct caption.

**Image generation issues.** While the text-to-image model may make occasional mistakes (such as generating the wrong object for unambiguous prompts), subtle errors can also arise from the interplay between the model and its prompt. The prompt may be ambiguous, such as using words that have multiple meanings (e.g., a *"walking stick"* can be both a cane or an insect), or may describe multiple objects with complex relationships that exacerbate mistakes (see Fig. 5(d)).

**Out-of-distribution sampling.** Ensuring that images sampled from an off-the-shelf generative model are part of the intended distribution (e.g., resembling IMAGENET) is difficult. We start our prompts with *"a realistic photograph"* in a bid to help steer the approximated distribution $\hat{p}(\mathbf{x}|y, \boldsymbol{z})$ away from artistic drawings and closer to the true distribution $p(\mathbf{x}|y, \boldsymbol{z})$. This approach is effective, but not always successful (see Fig. 5(e)). In some cases, finding a suitable prompt is not obvious (e.g., to output images from a particular medical domain; Kather et al., 2022) and fine-tuning models on the dataset of interest may be necessary.

**Privacy.** As we are generating large amounts of data, it is important to consider the associated privacy risks. While these risks can be mitigated by using generative models trained on public, non-sensitive data, more research on private generative modelling is necessary (Harder et al., 2022).

Despite these challenges, we foresee that large-scale generative models will increasingly be used as debugging tools. In this work, we introduced an automated pipeline that discovers failure cases in vision models. It constitutes a proof-of-concept that such a system allows for large-scale investigations of vision models in an open-ended manner.

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
