# OpenReview forum: "Discovering Bugs in Vision Models using Off-the-shelf Image Generation and Captioning"
_ICLR.cc/2023/Conference — Submitted to ICLR 2023_

### Official Review · Reviewer_v5Fg · 2022-10-24

**Confidence:** 3
**Correctness:** 2
**Technical Novelty And Significance:** 3
**Empirical Novelty And Significance:** 2
**Recommendation:** 3

**Clarity, Quality, Novelty And Reproducibility:**

Clarity Quality Novelty and Reproducibility
Clarity and quality are good while novelty and reproducibility are not good enough.

**Strength And Weaknesses:**

Strength: Automatically discovering failures is an interesting problem; The idea of applying image-to-text generative models to discover failures and obtain human-interpretable explanation of failure is novel.

Weaknesses:
The failure cases in this paper are all synthetic and there is still a large margin gap remaining between real-world data and data generated by generative models; Though a lot of well-trained generative models can be applied in the method, the costs of training them should not be ignored; The limitations mentioned in the paper are difficult to solve and will affect the effectiveness of the work. Due to the randomness in the process of selecting cases via generative models, it is difficult to prove that this framework can be reproduced and effective.

**Summary Of The Paper:**

This paper contributes a method to automatically find failure cases of image classifiers in an  oepn-ended manner by leverageing off-the-shelf, large-scale generative models.

**Summary Of The Review:**

This paper proposes a method to automatically find failure cases of image classifiers. The applying of generative models is novel, but the limitations mentioned in the paper are difficult to solve and will affect the effectiveness of the work.

---

> ### Author Response · Authors · 2022-11-11
> **Response to Reviewer v5Fg (1/2)**
>
> We appreciate that the reviewer finds this work novel and interesting. We have addressed each of the concerns below:
>
> *(1) The failure cases in this paper are all synthetic and there is still a large margin gap remaining between real-world data and data generated by generative models.*
>
> Yes, leveraging synthetic data to find failure cases is the main contribution of this paper. We demonstrate that generative models can be used to discover failures in vision classifiers. In Section 4.1, we show that discovered failures generalize to real images: the same failures arise if we use the captions as queries to Google Image Search and evaluate the classifier on the returned images.
>
> *(2) Though a lot of well-trained generative models can be applied in the method, the costs of training them should not be ignored.*
>
> It is true that training large-scale generative models is expensive. However, that cost can be amortized across evaluations. In this work, we use off-the-shelf generative models that were not trained or fine-tuned on the ImageNet domain. Furthermore, the failures we discover seem to generalize across various text-to-image models (see Section 4.1: Generalizability of failure descriptions). This indicates that the approach is general and extends to widely available text-to-image models (such as Stable Diffusion [Rombach et al., 2021](https://arxiv.org/abs/2112.10752)).
>
> That being said, even though our work purposefully uses off-the-shelf models, these large-scale models can be cheaply applied to new domains by fine-tuning [Ruiz et al., 2022](https://arxiv.org/abs/2208.12242) and do not necessarily need to be retrained from scratch to specialize (e.g. to medical domains [Kather et al., 2022](https://www.nature.com/articles/s41746-022-00634-5)).
>
> *(3) The limitations mentioned in the paper are difficult to solve and will affect the effectiveness of the work.*
>
> The fact that these challenges and limitations are hard to solve is not a reason to dismiss our work, which serves as an initial stepping stone towards building better "debuggers" for vision classifiers. While we foresee how to solve some of these challenges, others will require significantly more research. Moreover, solving or attempting to solve all of these challenges could be rather seen as a research agenda following up our work, therefore it is out of the scope and unreasonable in an 8 page paper.
>
> The main contribution of this paper is to demonstrate that we can leverage imperfect (yet good) generative models to debug classifier models today. We can uncover expected biases (jeep → snowplow) but also interesting new ones that are challenging to discover through other means (e.g. jellyfish → torch or seal → killer whale).

---

> > ### Comment · Reviewer_v5Fg · 2022-11-25
> > **The response to the rebuttal.**
> >
> > Thanks for your response.
> > For the first concern, what I want to express is that there is a big margin between the generative model and real images. In many cases, we cannot guarantee the reliability of the generated image. Deep learning, itself is a black box. It is difficult for you to use a black-box model to explain why another black-box model is making mistakes. The results are hardly convincing.
> > For the third concern, in the process of reading the literature, I have almost the same concerns about method limitations as expressed by the author. I hope that these limitations can be completely or at least partially resolved to ensure the effectiveness of the method.
> > I accept the explanation about the second and fourth concerns, but I think the first and third issues are more important, the author can continue to improve the model according to the limitations stated in the paper and the first issue I mentioned.

---

> > > ### Author Response · Authors · 2022-11-25
> > > **Response to Reviewer v5Fg (1 / 2)**
> > >
> > > Thank you for your response. Please see our answers to the points raised below.
> > >
> > > *Deep learning, itself is a black box. It is difficult for you to use a black-box model to explain why another black-box model is making mistakes.*
> > >
> > > We are not the first to use a machine learning model to investigate another machine learning model. This strategy has been used by a number of other papers. As a result, we remain optimistic that probing a machine learning model with other machine learning models can be a powerful and effective tool.
> > >
> > > Listed below are examples of prior work:
> > > 1. "Red Teaming Language Models with Language Models" [[Perez et al., 2022](https://arxiv.org/abs/2202.03286)] in which two machine learning models (a language model and a toxicity classifier) are used to investigate whether another large scale language model is prone to output toxic content.
> > > 2. "ImageNet-trained CNNs are biased towards texture; increasing shape bias improves accuracy and robustness" [[Geirhos et al., 2018](https://arxiv.org/abs/1811.12231)] in which a machine learning model is used to modify the texture of images to investigate whether classifier models are biased towards texture cues.
> > > 3. "Easily Accessible Text-to-Image Generation Amplifies Demographic Stereotypes at Large Scale" [[Bianchi et al., 2022](https://arxiv.org/abs/2211.03759)] in which CLIP embeddings are used to determine the properties of generated images at scale in order to determine whether generative models are biased.
> > >
> > > *The results are hardly convincing.*
> > >
> > > We invite the reviewer to provide stronger evidence as to why the results are not convincing. We believe that our work demonstrates without a doubt that generative models can be leveraged to discover failure cases of vision classifiers:
> > > 1. **We can find failures despite the text-to-image and image-to-text models being imperfect**: Our experimental work demonstrates that despite the fact that both models (text-to-image and image-to-text) are imperfect, we can still use them to discover interesting biases (see Sections 4.1 and B.1 in the supplementary).
> > > 2. **The same failures can be discovered on different classifier models** ranging from convolutional networks to transformer-based architectures (See Section B.1 in the supplementary).
> > > 3. **The failures generalise to real images and other generative models.** We demonstrate that the failures generalise to images downloaded from the web in Section 4.1 (Generalizability of failure descriptions). We further demonstrate that the failures generalise to Dall-E, Stable Diffusion. These failures are *not* random failures of the given generative models but generalise across generative models and real images.
> > > 4. **The discovered failures transfer well to other classifier models.** Failures found on a ResNet-50 transfer at a rate of 34% to a strong ViT-B/8 ImageNet model (pretrained on ImageNet-21K) despite not targeting that particular mode which has a baseline top-3 error rate of 4% (see sections 4.2 and B.2).
> > >
> > > In summary, we want to emphasize that our approach is akin to a test fuzzing software: it can be used to expose issues of biases and systematic failures, but the final decision rests with human practitioners. Overall, the main contribution of this paper is to demonstrate that we can leverage imperfect (yet good) generative models to debug classifier models today. We can uncover expected biases (jeep → snowplow) but also interesting new ones (e.g. jellyfish → torch or seal → killer whale) and explore biases in the training set as a result.

---

> > > ### Author Response · Authors · 2022-11-25
> > > **Response to Reviewer v5Fg (2 / 2)**
> > >
> > > *For the first concern, what I want to express is that there is a big margin between the generative model and real images. In many cases, we cannot guarantee the reliability of the generated image.*
> > >
> > > While generative models are imperfect, we find, quantitatively, that the generated images in general capture the class label and are more similar to ImageNet than other standard benchmarks for evaluating models trained on ImageNet.
> > >
> > > 1. **The generated images are more similar to ImageNet than those from ImageNet-A**. Models are evaluated under distribution shifts all the time to understand how they fail. For example, ImageNet-A evaluates models on images that are difficult for a ResNet-50 trained on ImageNet. Our dataset of failures is actually *more* similar to ImageNet than ImageNet-A (see Table 2) so while there is a domain gap between the generated and real images, it is no larger (and indeed smaller) than that of standard benchmarks people are already using.
> > > 2. **The generative model is outputting realistic images of the given class**. The generative model is capturing the class and caption in the vast majority of cases. For completeness, we measured the error rate of our text-to-image model on all 200 labels present in ImageNet-A by generating 10 images per class (totalling 2000 images). We note that in our pipeline we only consider egregious errors as wrong (where the classifier top-3 does not include the correct label or any label under the same WordNet parent). Of the generated images, 3.95% did not represent the correct label and 1.45% showed an item from the wrong WordNet family (e.g., asking for an ocarina sometimes generated a maraca - both are musical instruments but only the ocarina is a wind instrument). Only a single class (porcupine) was systematically misrepresented.
> > >
> > > We also remind the reviewer that **human curated datasets are also imperfect**. Images scraped from the web can also be erroneously labeled [[Vasudevan et al., 2022](https://arxiv.org/pdf/2205.04596.pdf), [Northcutt et al., 2021](https://arxiv.org/abs/2103.14749)]. However, we still use such images to train and test models. Moreover, our discussion in the first part of this response (1/2) on why the results **are** convincing demonstrates that these synthetic images are indeed capturing failure cases of the classifier.
> > >
> > > *For the third concern, in the process of reading the literature, I have almost the same concerns about method limitations as expressed by the author.*
> > >
> > > We are glad that the section discussing the limitations of our approach is useful and appropriate. This section is an important component of our work and should be considered as one of our contributions.
> > >
> > > While we have been clear and upfront about the limitations of our work, this is not a reason to dismiss our work provided it has demonstrated what it has set out to do.
> > > Our aim, as stated in the introduction, is to demonstrate how **large-scale, off-the-shelf, generative models can be combined to automate the discovery of bugs in machine learning models and produce compelling, interpretable descriptions of model failures.** We have demonstrated this for the following reasons:
> > > 1. We can leverage imperfect (yet good) generative models to debug classifier models today. We can uncover expected biases (jeep → snowplow) but also interesting new ones (e.g. jellyfish → torch or seal → killer whale).
> > > 2. We demonstrate that these failures transfer to different classifiers and generalise to other generative models and real images.
> > > 3. We can scale this approach to create datasets of failures and compare models.
> > > 4. We can also explore which images in the training set may be responsible for the failure.

---

> > > ### Author Response · Authors · 2022-12-06
> > > **Further results demonstrating our discovered failures transfer to real images**
> > >
> > > As there were concerns that the synthetic data would not find failures that generalise to real images, we ran a larger scale experiment on real images than that in 4.1. In this experiment, we take the original prompt and discovered prompt, and use them as queries to Google Image Search to download approximately 100 images (downloading more images leads to images that do not match the query). We then run several ResNet50 classifiers on these images to determine if:
> > > 1. The failure rate increases under the modified prompt.
> > > 2. The confusion rate for the target class increases under the modified prompt.
> > >
> > > We run this experiment for five ResNet50 classifiers initialised with different seeds. A result is considered a failure if its top-1 prediction is not in the same WordNet hierarchy (when considering two parents) as the true label. It is considered a confusion if the top-1 prediction is the same as the target class.  Results are reported in the table below for three failures we found automatically with our approach.
> > >
> > > |   Caption   		| Target class | Failure Rate on the ~100 images | Confusion rate for target class |
> > > | ----------------- | ---------------- |--------------------| ---- |
> > > | “a realistic photograph of a robin (oscine).”    | hummingbird | 13.4% +/- 2.4 | 1.1% +/- 1.1 |
> > > | “--.-- it is flying”    | hummingbird | 34.0% +/- 5.5 | 10.3% +/- 3.9 |
> > > | “a realistic photograph of a scorpion (anthropod).”    | crayfish | 10.7% +/- 1.4 | 0.2% +/- 0.5 |
> > > | “--.-- it is on a person’s hand”    | crayfish | 22.0% +/- 1.7 | 0.7% +/- 1.0 |
> > > | “a realistic photograph of an african chameleon (lizard).”    | agama | 3.0% +/- 0.5 | 1.7% +/- 0.6 |
> > > | “--.-- he is holding a stick. the chameleon is orange and white.”    | agama | 3.5% +/- 0.9 | 4.4% +/- 0.8 |
> > >
> > > As is clear, both properties hold (as they did for the *fly on a flower* and *crayfish in a net* examples presented in section 4.1): the overall failure rate and the confusion rate for the target class increase significantly. Thus providing further evidence that discovered failures generalise to real images and that synthetic images are able to discover compelling failures.
> > >
> > > We hope that this additional analysis convinces the reviewer that our proposed method is general and scalable. We point out that the above analysis takes a significant amount of time as it requires manually querying Google Image Search and manually checking each image (which further demonstrates why our proposed method is useful).

---

> ### Author Response · Authors · 2022-11-11
> **Response to Reviewer v5Fg (2/2)**
>
> *(4) Due to the randomness in the process of selecting cases via generative models, it is difficult to prove that this framework can be reproduced and effective.*
>
> First, grounding failures to language helps with generalization and reproducibility (as shown in Section 4.1). Second, failures are discovered by measuring the failure rates over many samples (we added more information on the number of samples we generate in the footnote in Table 1's caption).
>
> More specifically, we observe that:
>   1. failures found with one text-to-image model generalise to other text-to-image models (such as Dall-E 2 and Stable Diffusion) and real images (such as results from Google Image Search);
>   2. we can find the same failure across different vision classifiers. We have added a table in Appendix B (Table 4) which shows that ViTs also exhibit the same open-ended failures. The table is reproduced below and considers the open-ended failure case related to the fly and bee:
>
> | Model | Failure rate for baseline (towards bee) | Failure rate for discovered caption (towards bee) |
> |---|---|---|
> | ViT-B32 | 0.0002% | 0.02278% |
> | ViT-B8 | 0.0002% | 0.0027% |
>
> In both cases the discovered caption was “... fly (insect). It is on a flower.”
>
> To further demonstrate that our results are significant, we ran, for two of the open-ended failure cases, the same experiment 10 times and used a statistical test to demonstrate the significance of our results. For each failure case, we generate samples until we either find 10 images that cause the classifier to mispredict the class towards the target class (using top-1 accuracy instead of the typical top-3 to allow our experiment to finish on time for the rebuttal) or find no misclassification towards the target class within 20K samples. We do this for the original caption and the automatically discovered caption which should induce significantly more failures. We report the failure rate for the original and modified captions. We then compute p-values using the Mann-Whitney U test at a significance level of 0.005 to determine if the differences in failure rates are statistically significant. In light of the amount of computational resources required to run the models employed in our work, we believe analyzing statistical significance across 10 repetitions is sufficient to demonstrate our approach is effective and can be reproduced.
>
> | Original caption  | Discovered caption  | Target class | Failure rate for baseline | Failure rate for discovered caption  | P-value |
> | --- | --- | ----------- | ----------- | --- |----------- |
> | “... fly (insect).”  |  “... fly (insect). It is on a flower.”    | bee | 0.00% $\pm$ 0.00 |  0.58% $\pm$ 0.15     | $0.00015$ |
> | “... crayfish (crustacean).”  |  “... crayfish (crustacean). It is in a net.”    | chainlink fence | 0.00% $\pm$ 0.00 |  2.09% $\pm$ 0.56     | $6.34 \cdot 10^{-5}$ |
>
>
> As the p-values ($0.00015$, $6.34 \cdot 10^{-5}$)  are lower than $0.005$, we find a *significant* result that images of the discovered caption (e.g. “... crayfish (crustacean). It is in a net.”) are more often misclassified for the target class (e.g. chainlink fence) than those from the original caption (e.g. “... crayfish (crustacean).”). These results provide additional evidence that the contributions within our work are reproducible and effective at discovering failure cases in vision models. We have added this additional analysis in Appendix B.1 (Table 5).

---

### Official Review · Reviewer_GhTH · 2022-10-27

**Confidence:** 4
**Clarity, Quality, Novelty And Reproducibility:** The paper is well organized, technica…
**Correctness:** 3
**Technical Novelty And Significance:** 4
**Empirical Novelty And Significance:** 3
**Recommendation:** 6

**Strength And Weaknesses:**

Overall, the idea of exploiting off-the-shelf generative and captioning models to discover failures of image classifiers trained in ImageNet is interesting and novel. I am also impressed by the extensive experimental results (including visualization analysis and variants effect analysis).

Nevertheless, I have several small issues:
- Although adequate experiments are conducted on ResNet-50 on ImageNet, I am curious to see experiments on a stronger network backbone such as ResNet-152. This lies in the concern that a stronger network may not be easily found failures using the proposed method. As a matter of fact, I’m mostly convinced by the provided results. But I think this experiments will make the conclusions more solid.
- Do the discovered failures can further improve the object classification model’s performance?


**Summary Of The Paper:**

The paper proposes utilizing off-the-shelf text-to-image and image-to-text models to discover vision models' failures automatically. The experimental results are delighting and convincing to some extent. This paper is also inspiring and potentially useful to improve the generalizability of vision models.

**Summary Of The Review:**

Currently, I lean to positive for its novel idea.

---

> ### Author Response · Authors · 2022-11-11
> **Response to Reviewer GhTH**
>
> We appreciate that the reviewer finds this work novel and inspiring and found the experimental results interesting. We have addressed the reviewers’ questions below.
>
> *(1) Although adequate experiments are run on ResNet-50, what would be the results on a stronger backbone (like ResNet-152)?*
>
> We used a ResNet-50 for discovering open-ended failures as it allows us to run more experiments. We also note that we measured top-3 error rates on ImageNet when finding failures, making this a challenging backbone, as the top-3 error rate of our ResNet-50 is 11%. (We have added information on the models’ top-3 error on ImageNet in the revised paper in Fig. 4 and 7). Better models such as the ViT-B/8 have top-3 error rates around 4%. Despite this difference, we are still able to obtain large failure rates (of 44%) for the ViT-B/8 as illustrated in Fig. 7.
>
> However, to demonstrate that our method can be applied to a stronger backbone, we demonstrate we  can recreate the fly → bee case study with ViTs. Overall, we need to sample more images to surface errors in stronger models, but are still able to discover failures in an open-ended manner. We can
> 1. discover the exact same failure (and same caption);
> 2. demonstrate that all models exhibit the same bias (i.e., a fly on a flower is more often confused for a bee).
>
> We have chosen to focus on ViTs as they are both stronger than the ResNet-50 and ResNet-152 variants and have a very different architecture. We have added a table in Appendix B (Table 4) and reproduce that table below:
>
> | Model | Failure rate for baseline (towards bee) | Failure rate for discovered caption (towards bee) |
> |---|---|---|
> | ViT-B32 | 0.0002% | 0.02278%
> | ViT-B8 | 0.0002% | 0.0027%
>
> In both cases the discovered caption was “... fly (insect). It is on a flower.”
>
> *(2) Can the discovered failures further improve the classifier’s performance?*
>
> This is an interesting question but it was not the focus of our work. Much like test fuzzing software and software debuggers, we propose here an approach to discover failures and test biases. These failures can be used to inform future data collection processes and improve classifiers as a result. However, our focus is not on improving the underlying classifier using these results, which we believe is not obvious and requires further research.
>
> In summary, much like automatic code correction, more research is needed to fully understand how to best use discovered failures to improve the underlying classifier. As hinted in Section 4.1, we could automatically issue Google Image Search queries to gather more real images, but we leave this thought to future work.

---

### Official Review · Reviewer_B78A · 2022-10-27

**Confidence:** 3
**Correctness:** 3
**Technical Novelty And Significance:** 3
**Empirical Novelty And Significance:** 3
**Recommendation:** 6

**Clarity, Quality, Novelty And Reproducibility:**

It is well written and clear enough for someone unfamiliar with the topic to easily follow. The method is technically sound for the most part. This work uses existing SOTA models but the way they combined them as a new debugging tool is fairly novel.

**Strength And Weaknesses:**

Strength

* Formulated the problem of finding failures of image classifier in a probabilistic framework and showed how one can use off-the-shelf image-to-text captioning model and text-to-image generative model to create an interpretable and easy-to-use debugging tool.
* Proposed approach is more interpretable and naturalistic than existing approaches and shows promise in generalizability.


Weakness / other comments

* “In particular, we use three shots, with each shot being an image and caption pair. Images are publicly available online and are manually described using between four to seven short sentences that describe the subject of the photograph and its physical position with respect to the camera, as well as the background or context in which the subject appears” → This indicates that the method requires human input with some expert knowledge which raises a question on scalability and potential bias for this process.
* How can identified failures help improving the actual model? Further discussion / analysis on how the discovered failures could be used would be useful in order to give a better idea to the practitioners who would want to use this system to improve their model.
* “For each label y, we manually select target labels  ŷ (snow leopard, bee and chainlink fence respectively)” → Why was this set of ŷ selected? What is the impact of the choice ŷ on failure discovery?



**Summary Of The Paper:**

This paper describes a pipeline that can be used to identify failures of image classifiers. Proposed approach utilizes SOTA generative models to enable interpretable failure discovery of a model by providing targeted realistic images. Experiments are well conducted to demonstrate the benefit of the approach. Discussions on its limitations and potential concerns are well considered in the paper as well.

**Summary Of The Review:**

This paper clearly describes their proposed method which is well designed and backed up with proper analysis and experiments. Although I have some doubts of the claimed scalability and how practical it would be if we were to use them for model improvement, as a proof-of-concept I think it falls above the acceptance bar.

---

> ### Author Response · Authors · 2022-11-11
> **Response to Reviewer B78A**
>
> We thank the reviewer for their review. We address further questions below
>
> *(1) Using the few shot prompt indicates that the method requires human input with some expert knowledge which raises a question on scalability and potential bias for this process.*
>
> The few-shot prompt is chosen once for all experiments. We purposely used 3 generic images from the web to construct the few-shot prompt for the image-to-text model. These images are not part of the ImageNet dataset. Expert knowledge, if needed, is only required to annotate 3 images. Compared to annotating a full test set, the cost is negligible. In the specific case of ImageNet, we could also have used a few images from an existing dataset like COCO but ultimately decided that taking 3 random images from the web required less expertise and demonstrated better our approach. We have added more context about this in appendix A in the paper.
>
> *(2) How can identified failures help improving the actual model?*
>
> We appreciate that the reviewer is eager to leverage this approach to improve the underlying classifier and agree that this is an interesting future research direction. However, much like test fuzzing software and software debuggers, we propose here an approach to discover failures and test biases. These failures could be used to inform future data collection processes and improve classifiers as a result. However, our focus is not on improving the underlying classifier using these results, which we believe is not obvious and requires a significant amount of further research.
>
> In summary, much like automatic code correction, more research is needed to fully understand how to best use discovered failures to improve the underlying classifier. As hinted in Section 4.1, we could automatically issue Google Image Search queries to gather more real images, but we leave this thought to future work.
>
> *(3) Why was this set of ŷ selected? What is the impact of the choice ŷ on failure discovery?*
>
> We selected the class pairs because we found them particularly interesting. They allow us to build a qualitative demonstration of the ability of the approach to uncover both expected (fly → bee) and unexpected (crayfish → chainlink fence) biases. We note that 36 additional pairs are shown in the appendix. For these pairs, the source class is selected at random from the 200 classes of ImageNetA and the target class is automatically discovered by our approach. In other words, while we chose $\hat{y}$ for the qualitative demonstration, we did not do so when using it for finding 36 additional failures in the appendix (this was done automatically), so the choice of $\hat{y}$ is not necessary and does not impact the ability of our pipeline to discover failures.

---

### Official Review · Reviewer_38rC · 2022-10-30

**Confidence:** 4
**Correctness:** 3
**Technical Novelty And Significance:** 3
**Empirical Novelty And Significance:** 3
**Recommendation:** 3

**Clarity, Quality, Novelty And Reproducibility:**

The paper is clearly written. The topic of incorporating large-scale text-to-image and image-to-text models for finding the failure cases of image classification models is novel. The authors provide details of the method for reproducing the results.


**Strength And Weaknesses:**

Strengths:
1. Incorporating large-scale text-to-image and image-to-text models for diagnosing classification models is an interesting topic. The authors provide some insightful explorations.
2. The proposed method can be used to diagnose image classification models and provide language explanations for the failure modes.

Weaknesses:
The major limitation is that the proposed approach assumes that the text-to-image and image-to-text models are perfect. However, if error happens in text-to-image synthesis or image captioning synthesis, then the results will not be reliable. For example, what if the text-to-image synthesis model generates an image of a bee with the caption "a realistic photograph of a fly"? The authors only discussed this issue in one paragraph in limitations, but I think this is an important issue that should be investigated deeper. How frequently do the text-to-image synthesis and image captioning models make mistakes, and how these mistakes will affect the results? How robust is the proposed approach?

**Summary Of The Paper:**

This paper proposes an approach to automatically discover the failure cases of vision models under real-world settings. Off-the-shelf image-to-text and text-to-image models are leveraged to find such failures. Firstly, a conditional text-to-image synthesis model generates synthetic data based on the ground-truth label. Then, a captioning model is used to describe misclassified inputs. Next, the descriptions generated by the captioning model are used to synthesize more images to test whether the specific description causes a higher failure rate. This pipeline can be used to find the failure modes with language explanations for the classification models.

**Summary Of The Review:**

My major concern is the robustness of the approach. If the text-to-image models and the image captioning models make mistakes, the results will be misleading. This approach assumes that the text-to-image and image captioning models are perfect, or at least the error rate is much smaller than the image classification model. But I suspect this assumption is not always true.

---

> ### Author Response · Authors · 2022-11-11
> **Response to Reviewer 38rC**
>
> We thank the reviewer for their review and appreciate that they find the method interesting and novel. We address questions and concerns below.
>
> *The major limitation is that the proposed approach assumes that the text-to-image and image-to-text models are perfect. How frequently do the text-to-image synthesis and image captioning models make mistakes, and how these mistakes will affect the results? How robust is the proposed approach?*
>
> Our paper introduces a novel approach to debug image classifiers. Just like software debuggers and test fuzzers, we expect our work to be used in conjunction with human experts and not alone.
> That being said, the proposed approach can consistently find interesting failure cases as:
>
> 1. **We measure only egregious errors**: We only measure egregious errors from the classifier (i.e., the classifier top-3 prediction should not include the correct label or any labels under the same WordNet parent). This provision gives more freedom to the text-to-image model as it can also generate related objects without overly affecting the error rate measure. Furthermore, since errors are egregious, the process of catching such mistakes by non-expert humans is easier and less error prone.
> 2. **The image-to-text model is not required to be perfect**: Indeed, we always force completions from the baseline prompt, which means that the captions always contain the correct label. Additionally, cases where the completion is incorrect (the text does not match the image) but the text-to-image remains faithful to that completion are not problematic: if the new completion results in higher failure rates of the tested classifier we can still surface this failure (otherwise we do not).
> 3. **We find interesting failures despite the text-to-image and image-to-text models being imperfect**: Our experimental work demonstrates that despite the fact that both models (text-to-image and image-to-text) are imperfect, we can still use them to discover interesting biases. Please take a look at Appendix B.1 for more open-ended failures.
> 4. **Even human curated datasets are imperfect**: Images scraped from the web can also be erroneously labeled [[Vasudevan et al., 2022](https://arxiv.org/pdf/2205.04596.pdf)], [[Northcutt et al., 2021](https://arxiv.org/abs/2103.14749)]. However, we still use such images to train and test models.
> 5. **Using a delta in failure rate filters classes where the text-to-image model is poor**: We emphasize that our approach always measures a delta between the hypothesis (discovered caption) and the baseline (baseline prompt) and we only surface failing captions when they yield significantly higher failure rates. Thus if, for a particular class, the text-to-image model is bad at generating images of that class, the original failure rate will already be high and so generating images that are slightly different (e.g., by adding a flower) may have minimal impact on the failure rate.
> 6. **Potential for further automatic filtering**: Finally, systematic errors from the text-to-image model (such as always drawing a "bee" when asking for a "fly") could be automatically caught by monitoring the expected classifier error rate (e.g., obtained on the usual ImageNet validation set) and comparing it with the error on the baseline prompts. In essence, it is possible to quickly assess whether the text-to-image model produces valid images for all labels.
>
> **Error rate of text-to-image model:** For completeness, we also measured the error rate of our text-to-image model on all 200 labels present in ImageNet-A by generating 10 images per class (totalling 2000 images). We reiterate that we only consider egregious errors as wrong (where the classifier top-3 does not include the correct label or any label under the same WordNet parent). Of the generated images, 3.95% did not represent the correct label and 1.45% showed an item from the wrong WordNet family (e.g., asking for an ocarina sometimes generated a maraca - both are musical instruments but only the ocarina is a wind instrument). Only a single class (porcupine) was systematically misrepresented.
>
> Again, we want to emphasize that our approach is akin to a test fuzzing software: it can be used to expose issues of biases and systematic failures, but the final decision rests with human practitioners. Overall, the main contribution of this paper is to demonstrate that we can leverage imperfect (yet good) generative models to debug classifier models today. We can uncover expected biases (jeep → snowplow) but also interesting new ones (e.g. jellyfish → torch or seal → killer whale) and explore biases in the training set as a result.

---

> > ### Comment · Reviewer_38rC · 2022-11-25
> > **Thanks for the response**
> >
> > I would like to thank the authors for the detailed response. However, I am still not convinced because it is unclear how well the proposed approach can discover bugs given that the generative models do not fully capture the real data distribution, and that the approach is not fully automatic. In addition, the authors also admit that the approach can only measure egregious errors. So I am not convinced of the contributions and potential impact of this paper.

---

> > > ### Author Response · Authors · 2022-11-25
> > > **Response to Reviewer 38rC**
> > >
> > > Thank you for your response. Please see our answers to your points raised below.
> > >
> > > *However, I am still not convinced because it is unclear how well the proposed approach can discover bugs given that the generative models do not fully capture the real data distribution*
> > >
> > > We have clearly demonstrated that we can find bugs in classifiers trained on ImageNet in Sections 4.1 and B.1 in the supplementary. We have demonstrated that these failures generalise to other generative models and real images downloaded from the web. Our method also discovers issues with large, pretrained classifiers such as the ViT-L/16 evaluated in Section B.2.
> > >
> > > We also note that **no** image evaluation benchmark (such as ImageNet-A, ImageNet-R, ImageNet itself) captures the full distribution of real data. In fact we would argue that a generative model, as it is trained on large amounts of data downloaded from the web, can do a better job than these other datasets. In comparison to ImageNet, our synthetic data is more similar than that of ImageNet-A as demonstrated in Table 2. So in the same way that ImageNet-A is relevant to investigate how ImageNet trained classifiers may fail, so is our synthetic data.
> > >
> > > *the approach is not fully automatic.*
> > >
> > > We are not sure what the reviewer is referring to. The approach **is** automatic.
> > >
> > > *In addition, the authors also admit that the approach can only measure egregious errors.*
> > >
> > > We are not sure why the reviewer thinks this is a problem. We measure egregious errors to ensure failure cases that we surface are problematic. Mistaking two breeds of dogs is not as problematic as mistaking a dog for a bee. These egregious errors are precisely the errors a model designer would want to fix first. Moreover, what actually **causes** the failure can be subtle (e.g.~the green background behind a persian cat in Figure 2). Maybe the reviewer can elaborate on what types of failures they think would be interesting to discover.

---

> > > ### Author Response · Authors · 2022-12-06
> > > **Further results demonstrating our discovered failures transfer to real images**
> > >
> > > As there were concerns that the synthetic data would not find failures that generalise to real images, we ran a larger scale experiment on real images than that in 4.1. In this experiment, we take the original prompt and discovered prompt, and use them as queries to Google Image Search to download approximately 100 images (downloading more images leads to images that do not match the query). We then run several ResNet50 classifiers on these images to determine if:
> > > 1. The failure rate increases under the modified prompt.
> > > 2. The confusion rate for the target class increases under the modified prompt.
> > >
> > > We run this experiment for five ResNet50 classifiers initialised with different seeds. A result is considered a failure if its top-1 prediction is not in the same WordNet hierarchy (when considering two parents) as the true label. It is considered a confusion if the top-1 prediction is the same as the target class.  Results are reported in the table below for three failures we found automatically with our approach.
> > >
> > > |   Caption   		| Target class | Failure Rate on the ~100 images | Confusion rate for target class |
> > > | ----------------- | ---------------- |--------------------| ---- |
> > > | “a realistic photograph of a robin (oscine).”    | hummingbird | 13.4% +/- 2.4 | 1.1% +/- 1.1 |
> > > | “--.-- it is flying”    | hummingbird | 34.0% +/- 5.5 | 10.3% +/- 3.9 |
> > > | “a realistic photograph of a scorpion (anthropod).”    | crayfish | 10.7% +/- 1.4 | 0.2% +/- 0.5 |
> > > | “--.-- it is on a person’s hand”    | crayfish | 22.0% +/- 1.7 | 0.7% +/- 1.0 |
> > > | “a realistic photograph of an african chameleon (lizard).”    | agama | 3.0% +/- 0.5 | 1.7% +/- 0.6 |
> > > | “--.-- he is holding a stick. the chameleon is orange and white.”    | agama | 3.5% +/- 0.9 | 4.4% +/- 0.8 |
> > >
> > > As is clear, both properties hold (as they did for the *fly on a flower* and *crayfish in a net* examples presented in section 4.1): the overall failure rate and the confusion rate for the target class increase significantly. Thus providing further evidence that discovered failures generalise to real images and that synthetic images are able to discover compelling failures.
> > >
> > > We hope that this additional analysis convinces the reviewer that our proposed method is general and scalable. We point out that the above analysis takes a significant amount of time as it requires manually querying Google Image Search and manually checking each image (which further demonstrates why our proposed method is useful).

---

### Decision · Program_Chairs · 2023-01-20

**Decision:**

Reject

**Justification For Why Not Higher Score:**

Two of the reviewers had major concerns about the robustness of the approach, as the proposed approach kind of assumes that the text-to-image and image-to-text models are perfect (or, performs very well). After rebuttal, one reviewer even downgraded the score to 3.

**Justification For Why Not Lower Score:**

N/A

**Metareview: Summary, Strengths And Weaknesses:**

This paper proposes an approach to automatically discover the failure cases of vision models via using off-the-shelf image-to-text and text-to-image models.

Initially, this paper received scores of 3566; after rebuttal, the scores have changed to 3366. On one hand, all the reviewers agree that the studied problem is interesting, and the proposed method is novel. On the other hand, the major concern shared by the two reviewers who gave a score of 3 is that the proposed approach kind of assumes that the text-to-image and image-to-text models are perfect (or, performs very well), leaving the robustness of the approach unclear and the results might be misleading. For this concern, the authors have provided detailed response; however, it is unfortunately not convincing enough for reviewers to increase the scores.

On balance, the AC decides to recommend rejection based on the current draft, and encourages the authors to further refine the paper and submit the paper to a future conference.